EMBO
Molecular Medicine

# Cerebrospinal fluid tau, neurogranin, and neurofilament light in Alzheimer's disease

Niklas Mattsson[1,2,*], Philip S Insel[1,3], Sebastian Palmqvist[1,2], Erik Portelius[4], Henrik Zetterberg[4,5], Michael Weiner[3], Kaj Blennow[4], Oskar Hansson[1,2,**], for the Alzheimer's Disease Neuroimaging Initiative[†]

## Abstract

Cerebrospinal fluid (CSF) tau (total tau, T-tau), neurofilament light (NFL), and neurogranin (Ng) are potential biomarkers for neurodegeneration in Alzheimer's disease (AD). It is unknown whether these biomarkers provide similar or complementary information in AD. We examined 93 patients with AD, 187 patients with mild cognitive impairment, and 109 controls. T-tau, Ng, and NFL were all predictors of AD diagnosis. Combinations improved the diagnostic accuracy (AUC 85.5% for T-tau, Ng, and NFL) compared to individual biomarkers (T-tau 80.8%; Ng 71.4%; NFL 77.7%). T-tau and Ng were highly correlated ($\rho = 0.79$, $P < 0.001$) and strongly associated with β-amyloid (Aβ) pathology, and with longitudinal deterioration in cognition and brain structure, primarily in people with Aβ pathology. NFL on the other hand was not associated with Aβ pathology and was associated with cognitive decline and brain atrophy independent of Aβ. T-tau, Ng, and NFL provide partly independent information about neuronal injury and may be combined to improve the diagnostic accuracy for AD. T-tau and Ng reflect Aβ-dependent neurodegeneration, while NFL reflects neurodegeneration independently of Aβ pathology.

**Keywords** Alzheimer's; biomarker; CSF; neurodegeneration
**Subject Categories** Biomarkers & Diagnostic Imaging; Neuroscience

See also: **CE Teunissen & L Parnetti** (October 2016)

## Introduction

Alzheimer's disease (AD) is a progressive disease characterized by accumulation of amyloid β (Aβ) and tau pathologies, neurodegeneration, and cognitive and functional decline. Biomarkers, including cerebrospinal fluid (CSF) measurements, may quantify AD-related brain changes *in vivo*, which has revolutionized research, clinical trial design, and clinical practice (Mattsson *et al*, 2015a). CSF total tau (T-tau) is a well-studied biomarker, which is increased in neuronal degeneration (Blennow *et al*, 2006) and in AD patients already in early clinical stages (Mattsson *et al*, 2009; Albert *et al*, 2011; McKhann *et al*, 2011; Sperling *et al*, 2011; Dubois *et al*, 2014). CSF neurogranin (Ng) and neurofilament light (NFL) are two other biomarkers that have recently been suggested to measure neurodegeneration in AD. Ng is a post-synaptic protein (Gerendasy & Sutcliffe, 1997) and a putative marker of synaptic loss in AD (Portelius *et al*, 2015), an event which may be closely linked to development of cognitive decline (Portelius *et al*, 2015). CSF Ng is increased in AD compared to other dementias (Janelidze *et al*, 2016; Wellington *et al*, 2016), already in early clinical stages (Kvartsberg *et al*, 2015; Tarawneh *et al*, 2016). CSF Ng is also associated with brain atrophy (Portelius *et al*, 2015; Tarawneh *et al*, 2016) and reduced brain glucose uptake (Portelius *et al*, 2015). NFL is a putative marker of subcortical large-caliber axonal degeneration, and increased CSF NFL has been linked to inflammatory diseases (Christensen *et al*, 2013) and frontotemporal lobe dementia (Petzold *et al*, 2007). But CSF NFL is also relevant in AD (Zetterberg *et al*, 2016) since AD not only involves loss of cortical structures but also white matter injury (Migliaccio *et al*, 2012) and disconnection of cortical and subcortical regions (Delbeuck *et al*, 2003).

It is unknown whether T-tau, Ng, and NFL provide independent information about AD and whether measuring Ng or NFL in addition to T-tau improves the diagnostic accuracy for AD. The aim of this study was therefore to compare T-tau, Ng, and NFL for AD diagnosis and test their associations with other AD hallmarks. Specifically, we tested the hypotheses that (i) combinations of T-tau, Ng, and NFL increase the diagnostic accuracy for AD versus controls (CN) and for progressive mild cognitive impairment (PMCI) versus stable MCI (SMCI); (ii) T-tau, Ng, and NFL have

1  Clinical Memory Research Unit, Department of Clinical Sciences Malmö, Lund University, Lund, Sweden
2  Department of Neurology, Skåne University Hospital, Lund, Sweden
3  Department of Radiology, University of California San Francisco, San Francisco, CA, USA
4  Clinical Neurochemistry Laboratory, Institute of Neuroscience and Physiology, The Sahlgrenska Academy at University of Gothenburg, Sahlgrenska University Hospital, Mölndal, Sweden
5  Department of Molecular Neuroscience, UCL Institute of Neurology, London, UK
  *Corresponding author. Tel: +46 72 575 9329; E-mail: niklas.mattsson@med.lu.se
  **Corresponding author. Tel: +46 40 33 50 36; E-mail: oskar.hansson@med.lu.se
  †Data used in preparation of this article were obtained from the Alzheimer's Disease Neuroimaging Initiative (ADNI) database (adni.loni.usc.edu). As such, the investigators within the ADNI contributed to the design and implementation of ADNI and/or provided data but did not participate in analysis or writing of this report. A complete listing of ADNI investigators can be found at: http://adni.loni.usc.edu/wp-content/uploads/how_to_apply/ADNI_Acknowledgement_List.pdf

different associations with Aβ pathology and with different clinical stages of AD; and (iii) T-tau, Ng, and NFL have different associations with other AD features, including cognitive decline, brain atrophy, brain hypometabolism, and white matter hyperintensities (WMH).

# Results

The study included 93 patients with AD, 187 patients with MCI, and 109 controls (Table 1). For some analyses, we contrasted PMCI ($N = 104$) versus SMCI ($N = 65$, Table 2).

## Demographics

T-tau and Ng were higher in females than in males (T-tau: median 90 ng/l versus 80 ng/l, $P = 0.036$; Ng: 467 ng/l versus 374 ng/l, $P = 0.0024$), while NFL was higher in males (7.22 versus 7.04 [log] ng/l, $P < 0.001$). T-tau and Ng were higher in $APOE$ ε4$^+$ than in $APOE$ ε4$^-$ participants (T-tau: 103 ng/l versus 69 ng/l, $P < 0.001$; Ng: 491 ng/l versus 343 ng/l, $P < 0.001$), while NFL did not differ by $APOE$ ε4 status (7.16 versus 7.09 [log] ng/l, $P = 0.15$). NFL was higher ($\rho = 0.32$, $P < 0.001$) and Ng was lower ($\rho = -0.13$, $P = 0.013$) in older people, while T-tau ($\rho = -0.0092$, $P = 0.86$) did not vary with age. Likewise, NFL was higher ($\rho = 0.10$, $P = 0.040$) and Ng was lower ($\rho = -0.13$, $P = 0.0089$) in people with higher

education, while T-tau had no correlation with education ($\rho = -0.072$, $P = 0.16$).

## Correlations between CSF T-tau, Ng, and NFL

T-tau, Ng, and NFL were all correlated (Fig 1A–C), and the correlation between T-tau and Ng was especially strong. T-tau, Ng, and NFL correlated negatively with Aβ42 (Fig 1D–F).

## Diagnostic accuracy of CSF T-tau, Ng, and NFL

We tested accuracies for AD dementia versus CN and for PMCI versus SMCI using logistic regression models (Table 3). T-tau, Ng, and NFL were all significant predictors of AD. For single predictors, T-tau had the highest accuracy, followed by NFL and Ng. The model with the highest accuracy used all biomarkers together (AUC 85.5%). For PMCI versus SMCI, T-tau and Ng, but not NFL, were significant individual predictors. No combination had higher AUC than the model using only T-tau. We also evaluated models adjusted for demographics (age, sex, and education) and demographics plus Aβ42 (Table EV1). These models had higher AUCs than the basic models without covariates, but the overall results and relationship between the different models were similar.

We noted that there was a special relationship between T-tau and Ng, so that when Ng was adjusted for T-tau, the effect of Ng on AD diagnosis often changed from positive to negative (for example, from $\beta = 0.85$ to $\beta = -0.56$, Table 1). When adjusting for other covariates, this inversed effect of Ng was statistically significant, indicating that low Ng was associated with AD when the models were adjusted for T-tau (Table EV1).

## Biomarker combinations for classification of patients and controls

We extracted classification tables from the logistic regression models generated above, based on a threshold of 50% for the predicted probability of the logistic regression models. The results are summarized in Table 4. T-tau correctly classified 59 of 93 AD and 90 of 109 CN. Adding Ng improved the classification of CN to 93 of 109, without changing the classification of AD. In contrast, adding NFL improved the classification of AD to 61 of 93 but worsened the classification of CN to 87 of 109. Adding both Ng and NFL improved the classification of both AD (66 of 93, a relative increase of 12% from 59 of 93) and only slightly affected the classification of CN (89 of 109 CN). When also adjusting for age, sex, and Aβ42, NFL had the highest correct classification rate (correctly classifying 79 of 93 AD and 93 of 109 CN), and there was no improvement when combining biomarkers. For PMCI versus SMCI, all models had high correct classification of PMCI and poor classification of SMCI. Adjusting for age, sex, and Aβ42 greatly improved classification of SMCI.

## CSF T-tau, Ng, and NFL across clinical diagnoses and Aβ pathology

For the next set of analyses, participants were grouped based on the combination of clinical diagnosis and Aβ pathology (CN Aβ$^-$,

### Table 1. Demographics.

|  | CN | MCI | AD | *P*-value |
|---|---|---|---|---|
| *N* | 109 | 187 | 93 | NA |
| Age (years) | 75.7 (5.2) | 74.5 (7.5) | 74.7 (8.0) | 0.33 |
| Sex (F/M, % F) | 54/55 (50%) | 62/125 (33%) | 41/52 (44%) | 0.015 |
| *APOE* ε4 (+/−, % +) | 26/83 (24%) | 100/87 (54%) | 65/28 (70%) | 0.001 |
| Education (years) | 15.8 (2.9) | 15.8 (3.0) | 15.1 (3.2) | 0.25 |

Continuous data are mean (standard deviation). *P*-values tested by Fisher's exact and Kruskal–Wallis test. CN, healthy controls; MCI, mild cognitive impairment; AD, Alzheimer's disease dementia.

### Table 2. MCI demographics.

|  | SMCI | PMCI | *P*-value |
|---|---|---|---|
| *N* | 65 | 104 | NA |
| Age | 73. 9 (7.4) | 74.5 (7.6) | 0.44 |
| Sex (F/M, % F) | 21/44 (32%) | 37/67 (36%) | 0.74 |
| *APOE* ε4 (+/−, % +) | 28/37 (43%) | 63/41 (61%) | 0.039 |
| Education (years) | 16.0 (3.0) | 15.9 (3.0) | 0.52 |
| Clinical follow-up (years) | 4.5 (2.4) | 4.4 (2.5) | 0.22 |

Continuous data are mean (standard deviation). Note that the total number differs from the complete MCI cohort. For the SMCI versus PMCI comparisons we only included SMCI subjects with at least 2 years of follow-up. *P*-values tested by Fisher's exact test and Mann–Whitney *U*-test. SMCI, stable mild cognitive impairment; PMCI, progressive mild cognitive impairment.

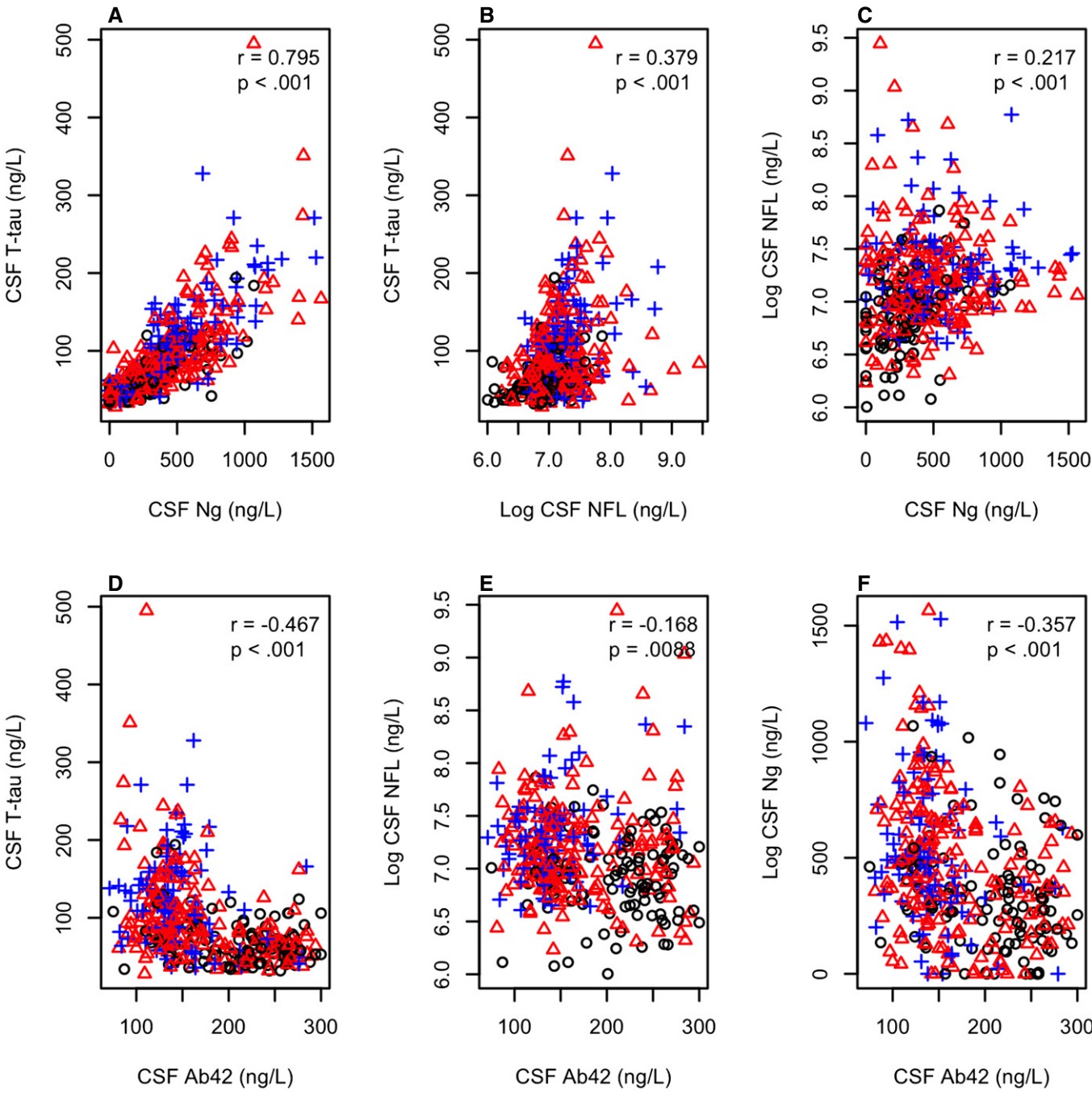

**Figure 1. Biomarker correlations.**

A–F  Associations between CSF T-tau, Ng, and NFL (panels A–C) and between these biomarkers and CSF Aβ42 (panels D–F). Black circles: CN (*n* = 109), red triangles: MCI (*n* = 187), blue crosses: AD (*n* = 93). Associations are shown for Spearman correlations. Aβ42 and T-tau were measured using the INNOBIA AlzBio3 kit (Fujirebio, Ghent, Belgium), Ng was measured using an in-house immunoassay for Ng (Portelius *et al*, 2015) and NFL was measured using the NF-light® ELISA kit (Uman Diagnostics, Umeå, Sweden).

$N = 69$; CN Aβ$^+$, $N = 40$; MCI Aβ$^-$, $N = 50$; MCI Aβ$^+$, $N = 137$; AD Aβ$^-$, $N = 8$; AD Aβ$^+$, $N = 85$). We performed different comparisons of CSF biomarkers between these groups. All models were adjusted for age and sex.

We first compared CSF biomarkers between Aβ-positive and Aβ-negative people within diagnosis (Fig 2 and upper part of

Table 5). Aβ positivity was associated with increased T-tau in all diagnostic groups and with increased Ng in MCI, but never with NFL.

We next compared CSF biomarkers between CN Aβ$^-$ and all other combinations of diagnosis and Aβ pathology. These comparisons were based on the theory that AD progresses from CN Aβ$^-$ to CN Aβ$^+$, MCI Aβ$^+$, and finally AD Aβ$^+$, while non-Aβ-dependent

**Table 3.    Diagnostic accuracy of CSF biomarkers.**

| Groups | Model | T-tau | Ng | NFL | AUC (95% CI) | AIC |
|---|---|---|---|---|---|---|
| AD versus CN | T-tau only | **1.61 (< 0.001)** | | | 80.8 (74.3–86.4) | 216.3 |
| | Ng only | | **0.85 (< 0.001)** | | 71.4[a] (64.1–77.9) | 254.7 |
| | NFL only | | | **1.38 (< 0.001)** | 77.7 (71.4–83.8) | 228.9 |
| | T-tau & Ng | **2.11 (< 0.001)** | −0.56 (0.074) | | 81.4[b] (75.2–87.3) | 215.0 |
| | T-tau & NFL | **1.33 (< 0.001)** | | **0.98 (< 0.001)** | 84.9[a,b,c] (79.8–89.8) | 198.1 |
| | Ng & NFL | | **0.61 (0.0016)** | **1.19 (< 0.001)** | 80.6[b,c,e] (74.7–86.2) | 219.7 |
| | T-tau & Ng & NFL | **1.80 (< 0.001)** | −0.52 (0.11) | **0.99 (< 0.001)** | 85.5[a–f] (80.6–90.6) | 197.4 |
| PMCI versus SMCI | T-tau only | **0.56 (0.0035)** | | | 67.7 (58.6–75.2) | 219.2 |
| | Ng only | | **0.39 (0.027)** | | 60.4[a] (50.6–68.8) | 223.9 |
| | NFL only | | | 0.32 (0.069) | 58.9 (48.8–68.0) | 225.6 |
| | T-tau & Ng | **0.60 (0.038)** | −0.046 (0.86) | | 68.2[b] (58.4–76.2) | 221.2 |
| | T-tau & NFL | **0.52 (0.0082)** | | 0.20 (0.23) | 67.5[b,c] (58.1–76.0) | 219.7 |
| | Ng & NFL | | **0.38 (0.29)** | 0.30 (0.076) | 63.0 (54.2–72.3) | 222.5 |
| | T-tau & Ng & NFL | 0.49 (0.10) | 0.028 (0.92) | 0.21 (0.23) | 67.3[b,c] (59.0–76.5) | 221.7 |

A separate logistic regression model was fit for each combination of neurodegeneration biomarkers in AD dementia versus CN and in PMCI versus SMCI. The table includes coefficients (log odds) with $P$-values, AUC, and AIC. For AUC, the letters a-f indicate significant differences ($P < 0.05$, tested by bootstrap) versus other models: T-tau (a), Ng (b), NFL (c), T-tau & Ng (d), T-tau & NFL (e), Ng & NFL (f). Bold values indicate significant associations. AD, Alzheimer's disease dementia; AIC, Akaike information criterion; AUC; area under the receiver operating characteristic curve; CN, healthy controls; PMCI, progressive mild cognitive impairment; SMCI, stable mild cognitive impairment.

**Table 4.    Classification tables.**

| Biomarkers | AD versus CN | | | PMCI versus SMCI | | |
|---|---|---|---|---|---|---|
| | Correct overall, % | Correct AD | Correct CN | Correct overall, % | Correct PMCI | Correct SMCI |
| No covariates | | | | | | |
| T-tau only | 73.8 | 59/93 | 90/109 | 68.0 | 95/104 | 20/65 |
| Ng only | 66.3 | 46/93 | 88/109 | 63.9 | 98/104 | 10/65 |
| NFL only | 72.3 | 60/93 | 86/109 | 64.5 | 102/104 | 7/65 |
| T-tau & Ng | 75.2 | 59/93 | 93/109 | 68.6 | 96/104 | 20/65 |
| T-tau & NFL | 73.3 | 61/93 | 87/109 | 69.8 | 96/104 | 22/65 |
| Ng & NFL | 70.8 | 59/93 | 84/109 | 65.1 | 95/104 | 15/65 |
| T-tau & Ng & NFL | 76.7 | 66/93 | 89/109 | 69.8 | 96/104 | 22/65 |
| Adjusted for age, sex, education, and Aβ42 | | | | | | |
| T-tau only | 81.7 | 73/93 | 92/109 | 72.7 | 90/104 | 33/65 |
| Ng only | 75.7 | 72/93 | 81/109 | 71.6 | 90/104 | 31/65 |
| NFL only | 85.1 | 79/93 | 93/109 | 73.4 | 93/104 | 31/65 |
| T-tau & Ng | 80.7 | 74/93 | 89/109 | 72.8 | 90/104 | 33/65 |
| T-tau & NFL | 84.7 | 78/93 | 93/109 | 73.4 | 93/104 | 31/65 |
| Ng & NFL | 84.7 | 79/93 | 92/109 | 71.0 | 89/104 | 31/65 |
| T-tau & Ng & NFL | 84.7 | 78/93 | 93/109 | 72.1 | 91/104 | 31/65 |

Classification tables from logistic regression models using a threshold of 50% for predicted probabilities. AD, Alzheimer's disease dementia; CN, healthy controls; PMCI, progressive mild cognitive impairment; SMCI, stable mild cognitive impairment.

cognitive decline may progress from CN Aβ− to MCI Aβ− and AD Aβ− (we consider these AD Aβ− cases to be clinically misdiagnosed). Compared to CN Aβ−, T-tau was increased in CN Aβ+, MCI Aβ+, and AD Aβ+, while Ng was increased in MCI Aβ+ and AD

Aβ+. In contrast, NFL was increased in all groups with cognitive decline (MCI Aβ+, AD Aβ+, MCI Aβ−, and AD Aβ−).

Finally, we compared the strengths of the associations between CSF biomarkers with different combinations of diagnosis

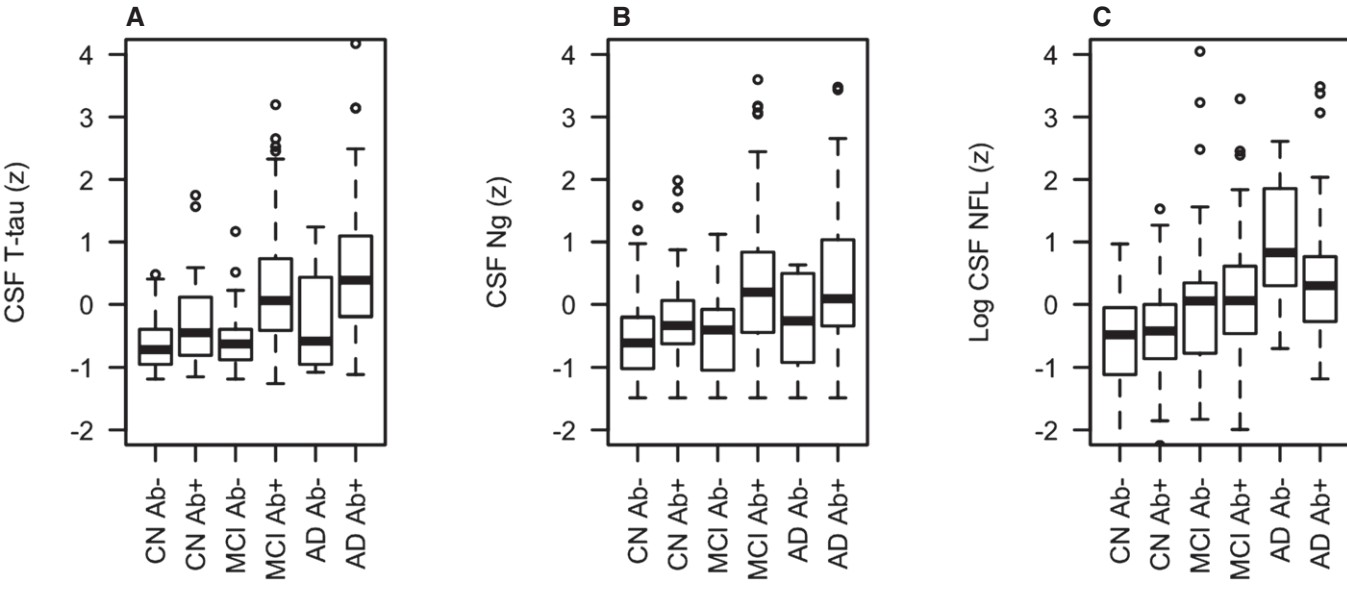

**Figure 2.  Biomarkers by diagnosis and amyloid pathology.**

A–C   CSF T-tau (panel A), Ng (panel B) and NFL (panel C) in different combinations of clinical diagnosis and Aβ pathology (CN Aβ⁻, $n$ = 69; CN Aβ⁺, $n$ = 40; MCI Aβ⁻, $n$ = 50; MCI Aβ⁺, $n$ = 137; AD Aβ⁻, $n$ = 8; AD Aβ⁺, $n$ = 85) (see Table 5 for comparisons between groups). Biomarker levels are standardized to $z$-scores and shown in box plots (indicating median and interquartile ranges; whiskers are defined as quartiles 1 and 3 $\pm$ 1.5 × interquartile range, respectively). Aβ42 and T-tau were measured using the INNOBIA AlzBio3 kit (Fujirebio, Ghent, Belgium), Ng was measured using an in-house immunoassay for Ng (Portelius et al, 2015), and NFL was measured using the NF-light® ELISA kit (Uman Diagnostics, Umeå, Sweden). Aβ was defined as CSF Aβ42 < 192 ng/l.

**Table 5.  Associations between biomarkers, clinical diagnosis and amyloid pathology.**

| Comparison | CSF T-tau | CSF Ng | CSF NFL | Difference T-tau versus Ng | Difference T-tau versus NFL | Difference Ng versus NFL |
|---|---|---|---|---|---|---|
| Associations between neurodegeneration biomarkers and Aβ pathology within diagnostic group | | | | | | |
| CN Aβ⁻ versus CN Aβ⁺ | **0.528 (0.0069)** | 0.332 (0.087) | 0.102 (0.60) | 0.197 (0.32) | **0.426 (0.031)** | 0.229 (0.24) |
| MCI Aβ⁻ versus MCI Aβ⁺ | **0.824 (< 0.001)** | **0.727 (< 0.001)** | 0.00778 (0.96) | 0.0970 (0.61) | **0.816 (< 0.001)** | **0.719 (0.00015)** |
| AD Aβ⁻ versus AD Aβ⁺ | **0.789 (0.040)** | 0.618 (0.11) | −0.632 (0.098) | 0.171 (0.67) | **1.420 (0.00057)** | **1.249 (0.0024)** |
| Associations between neurodegeneration biomarkers and combinations of clinical diagnosis and Aβ pathology | | | | | | |
| CN Aβ⁻ versus CN Aβ⁺ | **0.528 (0.0069)** | 0.332 (0.087) | 0.102 (0.60) | 0.197 (0.32) | **0.426 (0.031)** | 0.229 (0.24) |
| CN Aβ⁻ versus MCI Aβ⁺ | **0.916 (< 0.001)** | **0.816 (< 0.001)** | **0.798 (< 0.001)** | 0.100 (0.52) | 0.119 (0.45) | 0.0185 (0.91) |
| CN Aβ⁻ versus AD Aβ⁺ | **1.172 (< 0.001)** | **0.860 (< 0.001)** | **0.973 (< 0.001)** | **0.312 (0.045)** | 0.199 (0.20) | −0.113 (0.47) |
| CN Aβ⁻ versus MCI Aβ⁻ | 0.119 (0.51) | 0.120 (0.51) | **0.633 (0.00067)** | −0.00141 (0.99) | **−0.514 (0.014)** | **−0.513 (0.015)** |
| CN Aβ⁻ versus AD Aβ⁻ | 0.494 (0.18) | 0.0564 (0.88) | **1.40 (0.00028)** | 0.438 (0.24) | **−0.909 (0.016)** | **−1.35 (0.00040)** |

Results for comparisons between different combinations of clinical diagnosis and Aβ pathology (CN Aβ⁻, $n$ = 69; CN Aβ⁺, $n$ = 40; MCI Aβ⁻, $n$ = 50; MCI Aβ⁺, $n$ = 137; AD Aβ⁻, $n$ = 8; AD Aβ⁺, $n$ = 85) from linear mixed-effects models testing effects of Aβ within diagnostic groups (top 3 rows), and differences between Aβ⁻ CN and other combinations of Aβ and diagnosis (bottom 5 rows). The data correspond to Fig 2. Results are β-coefficient ($P$-value). For example, in the top row ("CN Aβ⁻ versus CN Aβ⁺"), the effect of T-tau indicates that CN Aβ⁺ was significantly associated with 0.528 standard deviations higher levels of T-tau. The columns marked "difference" test whether the comparison differed between two biomarkers. For example, in the top row ("CN Aβ⁻ versus CN Aβ⁺"), the difference for T-tau versus NFL indicates that CN Aβ⁺ was significantly more associated with T-tau than with NFL. Bold values indicate significant associations. CN, healthy controls; MCI, mild cognitive impairment; AD, Alzheimer's disease dementia.

and Aβ pathology. When comparing them head-to-head, T-tau and Ng had similar strengths of associations with all groups (shown in Table 5 as non-significant differences for T-tau and Ng). The only exception was that T-tau was more strongly associated with AD Aβ⁺ (i.e., the difference between CN Aβ⁻ and AD Aβ⁺ was larger for T-tau than for Ng; β = 0.312, $P$ = 0.045). NFL differed markedly from T-tau and Ng and was more strongly associated with MCI Aβ⁻ and AD Aβ⁻ than T-tau and Ng were (Fig 2 and Table 5).

### Associations with cognition, MRI, and FDG-PET measures

Finally, we tested associations between biomarkers and cognition, brain structure, brain metabolism, and WMH. Baseline data are

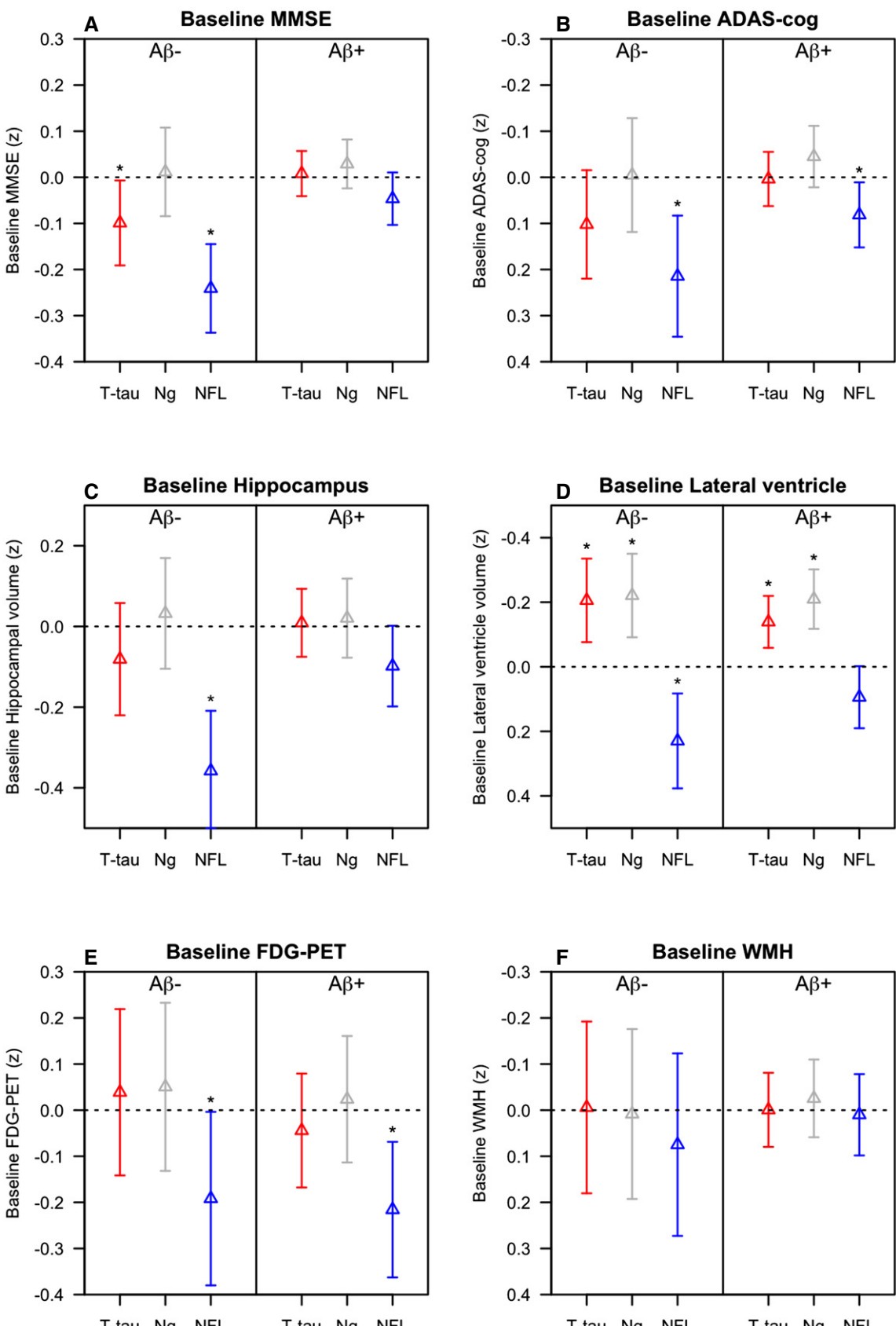

**Figure 3.**

**Figure 3.  Baseline associations between CSF T-tau, Ng, and NFL and other AD traits.**

A–F  Data are estimates (β-coefficients) from linear mixed-effects models, with 95% confidence intervals. The estimates are the main effects of the biomarkers, capturing the effects at study baseline. Effects were significant (*) for MMSE (A): T-tau ($P = 0.034$) and NFL ($P < 0.0001$) in $A\beta^-$ for ADAS-cog (B): NFL in $A\beta^-$ ($P = 0.0013$) and $A\beta^+$ ($P = 0.024$); for hippocampal volume (C): NFL in $A\beta^-$ ($P < 0.0001$); for lateral ventricles (D): T-tau ($P = 0.0019$), Ng ($P = 0.00092$), and NFL ($P = 0.0024$) in $A\beta^-$ and T-tau ($P = 0.00083$) and Ng ($P < 0.0001$) in $A\beta^+$ and for FDG-PET (E): NFL in $A\beta^-$ ($P = 0.046$) and $A\beta^+$ ($P = 0.0041$) people. Biomarkers and outcomes were standardized to $z$-scores. Note that the y-axes for ADAS-cog, ventricle size, and WMH are flipped, so that the lower ranges constantly reflect "worse" outcomes. Note also that the range of the y-axes differs for the different outcomes, for visualization purposes. Models were adjusted for age and sex, and education (for cognitive measures), and intracranial volume (for MRI measures). Aβ42 and T-tau were measured using the INNOBIA AlzBio3 kit (Fujirebio, Ghent, Belgium), Ng was measured using an in-house immunoassay for Ng (Portelius et al, 2015), and NFL was measured using the NF-light® ELISA kit (Uman Diagnostics, Umeå, Sweden).

shown in Fig 3, and longitudinal data are shown in Fig 4. The models were adjusted for age, sex, and education (for cognitive measures) and intracranial volume (for MRI measures).

At baseline, high T-tau was associated with poor cognition in $A\beta^-$. A detailed analysis suggested that this association was driven by the AD $A\beta^-$ individuals (i.e. subjects with a clinical diagnosis of AD but without evidence of Aβ pathology, not shown). T-tau was surprisingly associated with smaller ventricular volumes in both $A\beta^-$ and $A\beta^+$. Over time, high T-tau was associated with worsening cognition (MMSE in $A\beta^-$ and $A\beta^+$, ADAS-cog in $A\beta^+$), hippocampal atrophy, and hypometabolism (in $A\beta^+$). Ng was associated with smaller baseline ventricles in both $A\beta^-$ and $A\beta^+$, with worsening cognition, hippocampal atrophy, and hypometabolism over time in $A\beta^+$, and with slightly slower expansion of ventricle volume in $A\beta^-$.

To explore the unexpected associations between high T-tau and Ng and small baseline ventricles, we performed post hoc analyses within diagnostic subgroups (using ordinary linear regression, adjusted for intracranial volume, age, and sex). We found significant associations between high T-tau and small baseline ventricles in CN $A\beta^-$ (β = −0.25, $P = 0.016$) and MCI $A\beta^+$ (β = −0.18, $P = 0.0064$) and between high Ng and small baseline ventricles in MCI $A\beta^-$ (β = −0.39, $P = 0.0020$) and MCI $A\beta^+$ (β = −0.26, $P = 0.0012$). Even in groups where the associations were not significant, the highest T-tau and Ng levels were constantly seen in subjects with small baseline ventricles, and subjects with large baseline ventricles had low T-tau and Ng (Figs EV1 and EV2).

NFL had markedly different patterns of association compared to T-tau and Ng. At baseline, high NFL was associated with worse cognition, smaller hippocampal volumes, larger ventricles and lower FDG-PET in both $A\beta^-$ and $A\beta^+$ people (the effects were strongest in $A\beta^-$). Over time, NFL was associated with worsening cognition ($A\beta^-$ and $A\beta^+$), acceleration of hippocampal atrophy ($A\beta^-$), and expansion of ventricles ($A\beta^-$ and $A\beta^+$).

## Discussion

We compared CSF T-tau, NFL, and Ng, which are all putative biomarkers for neurodegeneration in AD. The main findings were that (i) combining biomarkers improved the diagnostic accuracy for AD versus CN compared to using individual biomarkers; (ii) T-tau and Ng were highly correlated and associated with Aβ pathology across clinical stages of AD, while NFL correlated with cognitive decline independent of Aβ pathology; and (iii) T-tau and Ng were associated with acceleration of cognitive decline, atrophy, and hypometabolism primarily in the presence of Aβ pathology, while NFL was associated with decline independent of Aβ pathology. In sum, our results support that CSF T-tau and Ng reflect neurodegeneration in AD, while NFL reflects neurodegeneration independent of AD. Combinations of these biomarkers provide partly complimentary information about diagnosis and pathology in people with cognitive decline.

The first main finding was that all biomarkers identified AD versus controls, and combinations improved the diagnostic accuracy compared to using individual biomarkers. The triple combination of T-tau, Ng, and NFL had the highest AUC, but the increase was small compared to the combination of T-tau and NFL. However, although combinations of biomarkers significantly increased the accuracy measured by AUC, the biomarker combinations tested here only had modest effects on the proportion of correctly classified people, especially when also adjusting the models for demographics and CSF Aβ42. For PMCI versus SMCI, T-tau and Ng, but not NFL, were significant predictors and no combination was significantly better than T-tau alone. We expected the biomarkers to be less accurate in MCI than in AD dementia, since some SMCI subjects may have non-AD diseases (e.g., vascular disease or non-AD tauopathies) reducing the specificity of the biomarkers.

The fact that NFL had high accuracy for AD is in line with several studies showing increased CSF levels of neurofilaments in AD

**Figure 4.  Longitudinal associations between CSF T-tau, Ng, and NFL and other AD traits.**

A–F  Data are estimates (β-coefficients) from linear mixed-effects models, with 95% confidence intervals. The estimates are the effect of time plus the biomarker by time interactions, capturing the longitudinal effects of the biomarkers. For each model, the "average" effect of time is also shown for comparison. Effects were significant (*), meaning that biomarker levels affected the slopes of the outcome, for MMSE (A): T-tau ($P = 0.021$) and NFL ($P < 0.0001$) in $A\beta^-$ and for T-tau ($P < 0.0001$), Ng ($P = 0.0031$) and NFL ($P < 0.0001$) in $A\beta^+$ for ADAS-cog (B): NFL in $A\beta^-$ ($P < 0.0001$) and for T-tau ($P < 0.0001$), Ng ($P = 0.0015$) and NFL ($P = 0.0027$) in $A\beta^+$ for hippocampal volume (C): NFL in $A\beta^-$ ($P = 0.0048$) and for T-tau ($P = 0.0015$) and Ng ($P = 0.0027$) in $A\beta^+$ for lateral ventricles (D): Ng ($P = 0.0051$) and NFL ($P < 0.0001$) in $A\beta^-$ and NFL ($P < 0.0001$) in $A\beta^+$; and for FDG-PET (E): T-tau ($P = 0.00038$) and Ng ($P = 0.0019$) in $A\beta^+$ people. Biomarkers and outcomes were standardized to $z$-scores. Note that the y-axes for ADAS-cog, ventricle size, and WMH are flipped, so that the lower ranges constantly reflect "worse" outcomes. Note also that the range of the y-axes differs for the different outcomes, for visualization purposes. Models were adjusted for age and sex, and education (for cognitive measures), and intracranial volume (for MRI measures). Aβ42 and T-tau were measured using the INNOBIA AlzBio3 kit (Fujirebio, Ghent, Belgium), Ng was measured using an in-house immunoassay for Ng (Portelius et al, 2015), and NFL was measured using the NF-light® ELISA kit (Uman Diagnostics, Umeå, Sweden).

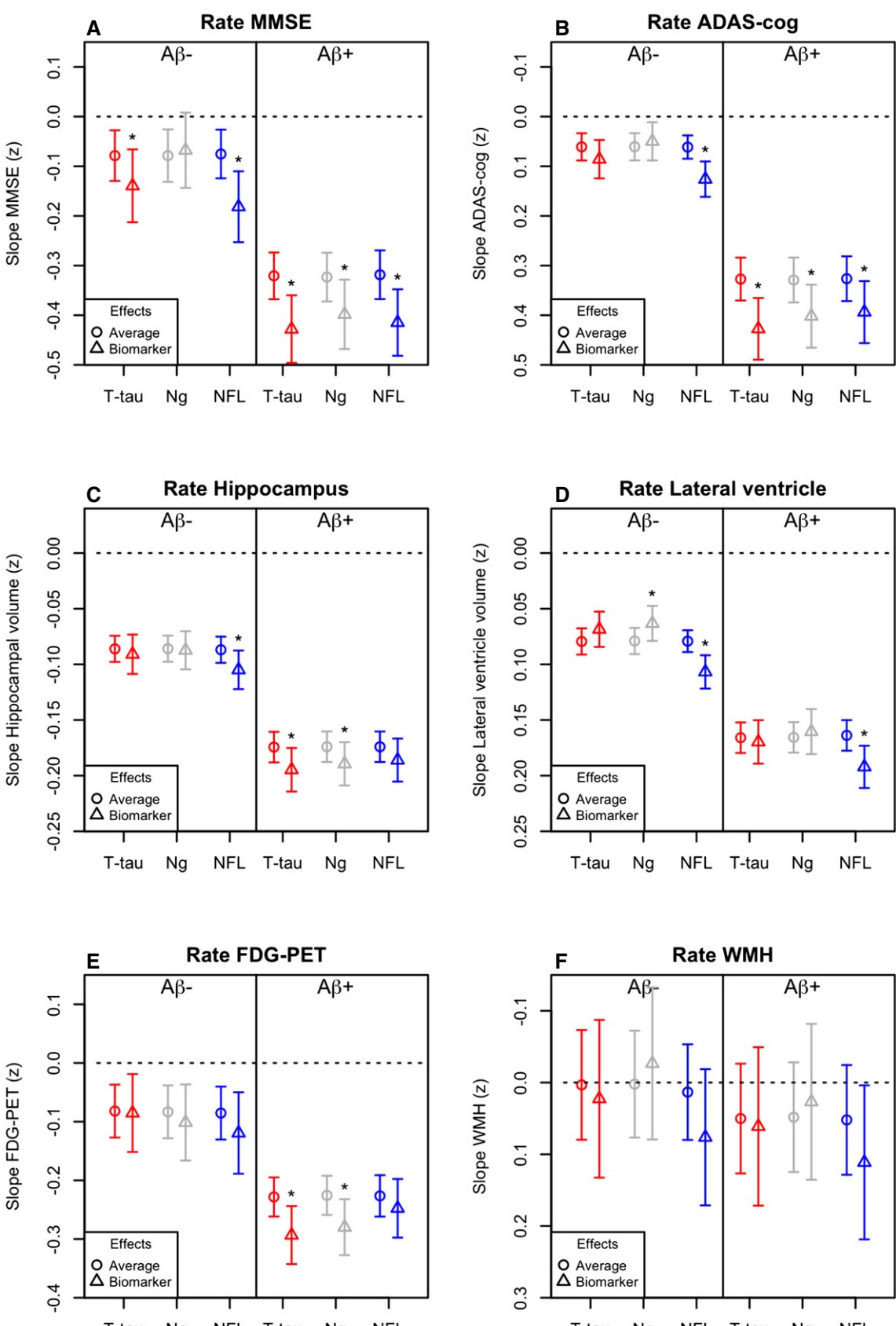

**Figure 4.**

(Rosengren *et al*, 1999; Sjögren *et al*, 2001; Hu *et al*, 2002; Zetterberg *et al*, 2016). However, NFL is also increased in other neurological conditions, including inflammatory diseases (Mattsson *et al*, 2010; Christensen *et al*, 2013), Creutzfeldt-Jakob disease (van Eijk *et al*, 2010), acute neuronal ischemia (Merisson *et al*, 2016), subcortical vascular dementia (Skillbäck *et al*, 2014), atypical parkinsonian disorders (Hall *et al*, 2012), and frontotemporal lobe dementia (FTD) (Petzold *et al*, 2007; Pijnenburg *et al*, 2007; Landqvist Waldo *et al*, 2013) where levels correlate with disease severity (Scherling *et al*, 2014). More work is needed to scrutinize CSF NFL in biomarker algorithms for differential diagnostics (de Jong *et al*, 2007).

The next main finding was that T-tau and Ng were strongly associated with Aβ pathology and more advanced clinical stages of AD. Specifically, T-tau and Ng were associated with Aβ pathology within diagnostic groups and with increased levels in CN Aβ$^+$, MCI Aβ$^+$, and AD Aβ$^+$ compared to CN Aβ$^-$. This links T-tau and Ng to Aβ-pathology and shows that this association is present at all clinical stages of AD, including the prodromal and preclinical stages (Dubois *et al*, 2016), which is in line with other recent reports on Ng in other cohorts (Hellwig *et al*, 2015; Tarawneh *et al*, 2016). In contrast, NFL was not associated with Aβ-pathology within diagnostic groups and was increased also in MCI Aβ$^-$ and AD Aβ$^-$ participants compared to CN Aβ$^-$. This shows that CSF NFL is a non-specific marker of injury, which is not primarily related to AD. This is consistent with previous reports on increased CSF NFL in different neurological conditions, as explained above.

The next finding was that the biomarkers had different relationships with different outcomes. T-tau and Ng primarily had associations with deterioration in the presence of Aβ pathology, which was seen in longitudinal analyses of cognition, hippocampal atrophy, and hypometabolism. In contrast, NFL had associations with cognition, ventricular expansion, and hypometabolism in both Aβ$^-$ and Aβ$^+$ people and with longitudinal hippocampal atrophy only in Aβ$^-$ people. Taken together, these results are compatible with a model where non-Aβ and non-tau pathologies (which may include vascular pathology, TDP-43, Lewy bodies and other pathologies) are associated with high NFL levels, expansion of ventricles, and some degree of hippocampal atrophy at baseline, but the progressive hippocampal volume loss seen in people harboring Aβ pathology is independent of NFL and is instead reflected by increased CSF levels of T-tau and Ng.

Unexpectedly, we found negative associations between T-tau and Ng and ventricle size at baseline. One possible interpretation of this is that T-tau and Ng not only reflect neuronal injury, but may also be related to normal neuronal function or transmission, hypothetically resulting in a relationship between high biomarker levels and small ventricle volume. Another unexpected finding was that when we combined T-tau and Ng in one model for AD diagnosis, the sign for Ng (but not T-tau) was reversed compared to when it was used alone (Table 3). This indicates that when Ng is used alone to predict AD, high levels are associated with disease, but when it is adjusted for T-tau, low levels are associated with disease. This finding fits with our speculation that Ng partly reflects normal neuronal function. More in-depth studies are needed to clarify mechanisms affecting CSF Ng levels in contrast to CSF T-tau. One recent study found that Ng increases over time in cognitively

normal people (Kester *et al*, 2015). The reason for this is unknown, but further longitudinal studies will be important to clarify the role of these biomarkers.

Several recent studies on Ng in AD have been published by us and other groups (Portelius *et al*, 2015; Tarawneh *et al*, 2016; Wellington *et al*, 2016). Compared to these, our study is novel by simultaneously investigating CSF T-tau, Ng, and NFL and by testing whether these biomarkers provide independent information about AD diagnosis and brain changes associated with AD in several different clinical stages of the disease. However, our study is not without limitations. We did not include non-AD neurodegenerative diseases, and we did not include neuropathology data. For the comparison between PMCI and SMCI, we only included SMCI subjects who were stable for at least 2 years to reduce the number of subjects erroneously classified as stable. However, it is possible that some SMCI subjects would have progressed to AD dementia with longer follow-up. A final limitation is that we used a fixed cut-point of CSF Aβ42 (192 ng/l) to test the effects of brain Aβ pathology, although it is possible that there may be subtle effects of emerging Aβ pathology in subjects with slightly higher CSF Aβ42 levels (Mattsson *et al*, 2014, 2015b; Insel *et al*, 2015, 2016).

T-tau, NFL, and Ng contribute independent information in characterization of CN, MCI, and AD. Combinations of these biomarkers, particularly T-tau and NFL, may increase the diagnostic accuracy of AD. T-tau and Ng are closely associated with Aβ pathology at all clinical stages of AD. The independent roles of Ng and T-tau need further clarification. In contrast to T-tau and Ng, NFL reflects neurodegeneration that is not associated with Aβ pathology. Future studies may explore whether CSF T-tau, Ng, and NFL respond differently to successful disease-modifying treatment against AD.

## Materials and Methods

### ADNI study design

Data were obtained from the Alzheimer's Disease Neuroimaging Initiative (ADNI) database (adni.loni.usc.edu). The principal investigator of this initiative is Michael W. Weiner, MD, VA Medical Center and University of California, San Francisco. ADNI subjects have been recruited from over 50 sites across the USA and Canada (for up-to-date information, see http://www.adni-info.org). Regional ethical committees of all institutions approved of the study. All subjects provided informed consent.

### ADNI subjects

We included all CN, MCI, and AD dementia subjects with available T-tau, NFL, and Ng data. Inclusion/exclusion criteria are described at http://www.adni-info.org. Briefly, all subjects included were between the ages of 55 and 90 years, had completed at least 6 years of education, were fluent in Spanish or English, and were free of any significant neurological disease other than AD. CN had Mini Mental State Examination (MMSE) score ≥ 24 and Clinical Dementia Rating (CDR) score 0. MCI had MMSE score ≥ 24, objective memory loss as shown on scores on delayed recall of the

Wechsler Memory Scale Logical Memory II (> 1 standard deviations below the normal mean), CDR 0.5, preserved activities of daily living, and absence of dementia. AD dementia patients fulfilled the National Institute of Neurological and Communicative Disorders and Stroke and the Alzheimer's Disease and Related Disorders Association (NINCDS-ADRDA) criteria for probable AD (McKhann *et al*, 1984) and had MMSE 20–26 and CDR 0.5–1.0. For some analyses, we contrasted PMCI versus SMCI. We defined PMCI as MCI subjects converting to AD dementia anytime during follow-up and SMCI as MCI subjects not converting to AD dementia during at least 2 year of follow-up.

### CSF measurements

CSF procedures have been described previously (Shaw *et al*, 2009). Aβ42, T-tau, and P-tau were measured at the ADNI biomarker core (University of Pennsylvania) using the multiplex xMAP Luminex platform (Luminex Corp, Austin, TX, USA) with the INNOBIA AlzBio3 kit (Fujirebio, Ghent, Belgium). Ng and NFL were measured at the Clinical Neurochemistry Laboratory at University of Gothenburg, Mölndal, Sweden, using an in-house immunoassay for Ng (Portelius *et al*, 2015) and a commercial ELISA for NFL (NF-light® ELISA, Uman Diagnostics, Umeå, Sweden) (Zetterberg *et al*, 2016). We excluded two subjects who were significant outliers in Ng measurements (CSF Ng > 2,000 ng/l).

### Cognition

Cognition was assessed by MMSE and ADAS-Cog11 up to 13 times: baseline and at 6, 12, 18, 24, 36, 48, 60, 72, 84, 96, 108, and 120 months after baseline.

### Brain structure

Structural magnetic resonance imaging brain scans were acquired using 1.5 Tesla MRI scanners up to nine times: baseline and at 6, 12, 18, 24, 36, 48, 60, and 72 months. We used a standardized protocol including T1-weighted MRI scans using a sagittal volumetric magnetization prepared rapid gradient echo (MP-RAGE) sequence (Jack *et al*, 2008). Automated volume measures were performed with FreeSurfer. We used averaged volume measurements for the right and left hippocampi and combined volumes for the lateral ventricles.

### Brain metabolism

FDG-PET image data were acquired up to 11 times (baseline and at 6, 12, 18, 24, 36, 48, 60, 72, 84, and 96 months after baseline) (Landau *et al*, 2012). We used mean FDG-PET counts of the lateral and medial frontal, anterior, and posterior cingulate, lateral parietal, and lateral temporal regions.

### White matter hyperintensities

White matter hyperintensities (WMH) were quantified up to seven times (baseline and at 6, 12, 18, 24, 36, and 48 months after baseline) using a fully automated protocol (Schwarz *et al*, 2009).

### Statistical analysis

Nonparametric tests (Kruskal–Wallis, Mann–Whitney *U*, and Spearman correlation) were used to test associations between biomarkers and demographic factors. The main analyses consisted of 4 different parts:

1    Diagnostic accuracies were tested in logistic regression models separately for AD versus CN and PMCI versus SMCI. All models were evaluated for significance of the included biomarkers, overall diagnostic accuracy (area under the receiver operator characteristics curve, AUC), and overall fit penalized for the number of predictors (Akaike information criterion, AIC). Differences between AUCs were calculated in a bootstrap procedure with resampling (*B* = 1,000 iterations). We also tested models adjusted for age, sex, education, and CSF Aβ42. We extracted classification tables from the logistic regression models to quantify correct classifications of AD, CN, PMCI, and SMCI, using a 50% threshold of predicted probability.

2    We next tested the effects of Aβ on biomarkers within diagnostic groups. We dichotomized each diagnostic group using a previously established cutoff of CSF Aβ42 (< 192 ng/l; Shaw *et al*, 2009) and compared Aβ⁻ versus Aβ⁺. CSF biomarkers were standardized and concatenated into a single response vector, which was used as the dependent variable in linear mixed-effects models. The independent variable was an interaction between Aβ and a factor for biomarker type, adjusted for both main effects, age, and sex. All models included a random intercept. The interaction between Aβ and biomarker type is an estimate of the different effect of Aβ in two different CSF biomarkers.

3    We next tested the combination of diagnostic group and Aβ status as predictor ("group"). We hypothesized that AD progresses from CN Aβ⁻ to CN Aβ⁺, MCI Aβ⁺, and AD Aβ⁺. In this paradigm, MCI Aβ⁻ and AD Aβ⁻ represent non-AD causes of cognitive decline (we consider the AD Aβ⁻ cases likely to be clinically misdiagnosed). We again used linear mixed-effects model with concatenated CSF biomarker levels as the response, and an interaction between "group" and a factor for biomarker type as predictor (with CN Aβ⁻ as the reference group), adjusted for age and sex.

4    Finally, we tested CSF biomarkers as predictors of different AD features, including MMSE, ADAS-cog, hippocampal volume, ventricular volume, FDG-PET, and WMH. This was done with longitudinal response data, using linear mixed-effects models. The predictors were a biomarker by time (years) interaction, age, sex, diagnosis (CN, MCI, and AD), and all main effects. Cognition was also adjusted for education and volume measures for intracranial volume. The models included random intercepts and slopes and an unstructured covariance matrix for the random effects. Models were tested separately for Aβ⁻ and Aβ⁺ people.

NFL was used after logarithmic transformation. All continuous variables were standardized (scaled and centered). β-coefficients from regressions refer to standardized effects (β = 1 indicates that a 1 standard deviation increase in the tested biomarker was associated with a 1 standard deviation increase in the dependent variable). We

checked model assumptions by inspecting residuals (correlations with fitted values and predictors of interest, histograms and q–q plots). All tests were two sided. Significance was determined at $P < 0.05$. All statistics were done using R (v. 3.2.3, The R Foundation for Statistical Computing).

**Expanded View** for this article is available online.

## Acknowledgements

Data collection and sharing for this project was funded by the Alzheimer's Disease Neuroimaging Initiative (ADNI) (National Institutes of Health Grant U01 AG024904) and DOD ADNI (Department of Defense award number W81XWH-12-2-0012). ADNI is funded by the National Institute on Aging, the National Institute of Biomedical Imaging and Bioengineering, and through generous contributions from the following: Alzheimer's Association; Alzheimer's Drug Discovery Foundation; Araclon Biotech; BioClinica, Inc.; Biogen Idec Inc.; Bristol-Myers Squibb Company; Eisai Inc.; Elan Pharmaceuticals, Inc.; Eli Lilly and Company; EuroImmun; F. Hoffmann-La Roche Ltd and its affiliated company Genentech, Inc.; Fujirebio; GE Healthcare; IXICO Ltd.; Janssen Alzheimer Immunotherapy Research & Development, LLC.; Johnson & Johnson Pharmaceutical Research & Development LLC.; Medpace, Inc.; Merck & Co., Inc.; Meso Scale Diagnostics, LLC.; NeuroRx Research; Neurotrack Technologies; Novartis Pharmaceuticals Corporation; Pfizer Inc.; Piramal Imaging; Servier; Synarc Inc.; and Takeda Pharmaceutical Company. The Canadian Institutes of Health Research is providing funds to support ADNI clinical sites in Canada. Private sector contributions are facilitated by the Foundation for the National Institutes of Health (www.fnih.org). The grantee organization is the Northern California Institute for Research and Education, and the study is coordinated by the Alzheimer's Disease Cooperative Study at the University of California, San Diego. ADNI data are disseminated by the Laboratory for Neuro Imaging at the University of Southern California.

This research was also supported by the European Research Council, the Swedish Research Council, the Strategic Research Area MultiPark (Multidisciplinary Research in Parkinson's disease) at Lund University, the Swedish Brain Foundation, the Skåne University Hospital Foundation, the Swedish Alzheimer Association, Stiftelsen för Gamla Tjänarinnor, the Swedish federal government under the ALF Agreement, the Thelma Zoéga Foundation, the Greta and Johan Kock Foundation, the Magnus Bergvall Foundation, the Knut and Alice Wallenberg Foundation, and the Torsten Söderberg Foundation.

NM had full access to all the data in the study and takes responsibility for the integrity of the data and the accuracy of the data analysis. No sponsor had any role in the design and conduct of the study; collection, management, analysis, and interpretation of the data; and preparation, review, or approval of the manuscript; and decision to submit the manuscript for publication.

## Author contributions

NM, KB, and OH involved in study conception and design. MW, EP, HZ, and KB participated in acquisition of data. NM, PI, and OH involved in analysis and interpretation of the data. NM drafted the manuscript. PI, SP, EP, HZ, MW, KB, and OH involved in critical revision of the manuscript.

## Conflict of interest

NM, SP, EP, and OH declare that they have no conflict of interest. HZ and KB are co-founders of Brain Biomarker Solutions in Gothenburg AB, a GU Venture-based platform company at the University of Gothenburg. MW has served on the Scientific Advisory Boards for Pfizer, Alzheon, Inc., Eli Lilly; has provided consulting to Synarc, Pfizer, Janssen, Alzheimer's Drug Discovery Foundation (ADDF), Avid Radiopharmaceuticals, Araclon, Merck, Biogen Idec, BioClinica, and Genentech; holds stock options with Alzheon, Inc; and has received funding for academic travel from Pfizer, Kenes, Intl.; UCSD; ADCS, University Center Hospital, Toulouse, Araclon, AC Immune, Nutricia, Eli Lilly, New York Academy of Sciences (NYAS), The Alzheimer's Association, Merck, Alzheimer's Drug Discovery Foundation (ADDF), Tokyo University, Kyoto University, Weill-Cornell University, Rockafeller University, Memorial Sloan-Kettering Cancer Center, and Biogen Idec.

## For more information

ADNI, http://adni.loni.usc.edu

## The paper explained

### Problem

Several cerebrospinal fluid biomarkers have been suggested to measure neurodegeneration in Alzheimer's disease. It is unclear to what degree these different biomarkers provide similar or complementary information.

### Results

Cerebrospinal fluid total tau, neurogranin, and neurofilament light represent different aspects of neurodegeneration in Alzheimer's disease. Combinations of these biomarkers improve the diagnostic accuracy compared to using individual biomarkers. Total tau and neurogranin are highly correlated and primarily correlate with degeneration in the presence of Aβ pathology. Neurofilament light correlates with degeneration independent of Aβ pathology.

### Impact

Total tau and neurogranin may be used to detect Aβ-dependent neurodegeneration, while neurofilament light may be used to monitor degeneration independent of Aβ pathology. This may have consequences for work-up in clinical practice and for enrichment and follow-up in clinical trials of Alzheimer's disease.

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
