## [Review Process File · EMBO Molecular Medicine]

Cerebrospinal fluid tau, neurogranin and neurofilament light in Alzheimer's disease

Niklas Mattsson, Philip Insel, Sebastian Palmqvist, Erik Portelius, Henrik Zetterberg, Michael Weiner, Kaj Blennow, Oskar Hansson, for the Alzheimer's Disease Neuroimaging Initiative

Corresponding author: Niklas Mattsson and Oskar Hansson, Lund University

Review timeline:	Submission date:	25 April 2016
	Editorial Decision:	31 May 2016
	Revision received:	22 June 2016
	Acceptance:	08 July 2016

Transaction Report:

Editor: Céline Carret

1st Editorial Decision

31 May 2016

Thank you for the submission of your manuscript to EMBO Molecular Medicine. We have now heard back from the two referees whom we asked to evaluate your manuscript.

You will see that both referees find the study to be of interest, even if referee 1 is rather succinct. Referee 2 however is more detailed and while supportive, suggests improving the clinical/translational relevance of the data, which in light of our scope, we want to insist upon.

We would welcome the submission of a revised version for further consideration and depending on the nature of the revisions, this may be sent back to the referees for another round of review. Please note that it is EMBO Molecular Medicine policy to allow only a single round of revision and that, as acceptance or rejection of the manuscript may depend on another round of review, your responses should be as complete as possible.

Revised manuscripts should be submitted within three months of a request for revision; they will otherwise be treated as new submissions, except under exceptional circumstances in which a short extension is obtained from the editor.

In order to gain time should your article be considered for publication, please carefully double check our author's guidelines and formatting details below prior to resubmitting your article.

I look forward to seeing a revised form of your manuscript as soon as possible.

***** Reviewer's comments *****

Referee #1 (Comments on Novelty/Model System):

This study allows one step further in the use of CSF biomarkers in clinical routine use, since it enlarges the panel and also links the results to the pathophysiology behind.

Referee #1 (Remarks):

Really good work and clearly written. The findings described are important.

Referee #2 (Remarks):

This is an important paper, and well performed study, showing the added value of analysing 3 different non-amyloid biomarkers in different clinical groups, in relation to amyloid positivity and negativity within the clinical groups, and the relation of the biomarkers to cognitive and MRI parameters stratified by amyloid positivity. So, very comprehensive and very clearly presented. The conclusions show that Ng does not have real added value to measuring Tau, and that NfL has added value. Also clearly shown is, due to the study design and statistical approach, that NfL is not related to amyloid pathology and thus provided other information.

I have a few suggestions for improvement, which can also be seen as discussion points. So, I would like to hear the authors answers.

Abstract: please add some results in the abstract where possible (e.g. AUCs).

Text in results section about abeta positivity could be structured a bit clearer: why compare abeta+ in AD with CNabeta-? It would help clarity if abeta+ or - is compared within diagnostic groups only, and not across diagnostic groups, since the significance of a difference in amyloid positive AD compared to CN negative is not clear to me. It reduces also the numbers of tests performed, increasing the strengths of the other results (not only in understanding but also statistically).

For all associations and group differences: given the correlations with age and sex, all associations must be corrected for these confounding factors. It is not clear from the methods if this has been done for statistical hypothesis number 3. I believe that the correction for age and sex should be stressed in the narrative of the results as well, and in the figure legends.

Since the aim is to define the added diagnostic value of combining the biomarkers, I expect also more discussion and some explanatory figures useful for the clinic, such as numbers of correctly classified patients: what is the increase in correct classification if neurofilament light is added to the diagnostic workup compared to amyloid beta and Tau alone? That will help translation to clinical practise.

The conclusion in 'paper explained' that tau and neurogranin can be used for monitoring is too hypothetical and not supported by the data, that do not include longitudinal sampling (only longitudinal clinical data).

Although I do acknowledge the tremendous and important contribution of the Gothenburg studies and of the first author to the development of the field, it would be good if external citations are used more. There are places where these would have been at least equally good papers (e.g. use of NfL for monitoring neuroinflammation; Ng papers of Fagan group).

AUC and AIC: abbreviations are not explained in the legend of table 3. Furthermore, it would be better to have confidence intervals for the AUCs, which allows the reader to judge better the significance of the increase in AUC in extended models.

Referee #1 (Comments on Novelty/Model System):

This study allows one step further in the use of CSF biomarkers in clinical routine use, since it enlarges the panel and also links the results to the pathophysiology behind.

Referee #1 (Remarks):

Really good work and clearly written. The findings described are important.

*** We appreciate these kind words about our paper.**

Referee #2 (Remarks):

This is an important paper, and well performed study, showing the added value of analysing 3 different non-amyloid biomarkers in different clinical groups, in relation to amyloid positivity and negativity within the clinical groups, and the relation of the biomarkers to cognitive and MRI parameters stratified by amyloid positivity. So, very comprehensive and very clearly presented. The conclusions show that Ng does not have real added value to measuring Tau, and that NfL has added value. Also clearly shown is, due to the study design and statistical approach, that NfL is not related to amyloid pathology and thus provided other information.

I have a few suggestions for improvement, which can also be seen as discussion points. So, I would like to hear the authors answers.

Abstract: please add some results in the abstract where possible (e.g. AUCs).

*** We have added results in the abstract (page 3).**

Text in results section about abeta positivity could be structured a bit clearer: why compare abeta+ in AD with CNabeta-? It would help clarity if abeta+ or - is compared within diagnostic groups only, and not across diagnostic groups, since the significance of a difference in amyloid positive AD compared to CN negative is not clear to me. It reduces also the numbers of tests performed, increasing the strengths of the other results (not only in understanding but also statistically).

*** We have restructured the section about Abeta (AB) positivity to make these results clearer (page 7, line 9 – page 8, line 8). We did the comparisons between CN AB- and CN AB+, MCI AB- /AB+, and AD AB-/AB+ because we wanted we examine if CSF injury biomarkers are differently related to different disease trajectories. As we explain in the methods section (page 16, lines 12-14), and as we now also mention in this paragraph in the results section, we believe that AD follows a sequence of events during its development in humans. First, CN AB- convert to AB+ without developing cognitive decline (CN AB+). Later, CN AB+ convert to MCI AB+ and finally AD AB+. In contrast, people who develop non-AB-dependent cognitive decline may also start as CN AB- but convert to MCI AB- and AD AB- (note that we consider these clinical “AD” diagnoses likely to be erroneous). We found that CSF injury biomarkers differentiate between these two trajectories as shown in Figure 2 and Table 5. For example, MCI AB+ have significantly higher CSF Ng, CSF T-tau and CSF NFL compared to CN AB-, while MCI AB- only have higher CSF NFL compared to CN AB-, and do not differ from CN AB- in CSF Ng or T-tau levels. This supports the idea that CSF Ng and CSF T-tau are associated with development of AD, while CSF NFL is associated with cognitive decline independent of AB.**

For all associations and group differences: given the correlations with age and sex, all associations must be corrected for these confounding factors. It is not clear from the methods if this has been done for statistical hypothesis number 3. I believe that the correction for age and sex should be stressed in the narrative of the results as well, and in the figure legends.

***Hypothesis 3 tests were also adjusted for age and sex. We now explain this in the statistical methods section (page 16, line 17), and in the narrative and in the figure legends.**

Since the aim is to define the added diagnostic value of combining the biomarkers, I expect also more discussion and some explanatory figures useful for the clinic, such as numbers of correctly classified patients: what is the increase in correct classification if neurofilament light is added to the diagnostic workup compared to amyloid beta and Tau alone? That will help translation to clinical practise.

***We have now included classification data generated from the logistic regression models (page 6, line 20 – page 7, line 7). The data are shown in a new table (Table 4) and provide numbers for classifications of AD, CN, PMCI and SMCI with different combinations of biomarkers, and when adjusting for Ab42 and demographic covariates. For the basic models, these results show a relative increase of 12% for classification of AD when combining biomarkers compared to when only using T-tau alone. In the more complex models the effects of combining biomarkers were smaller.**

The conclusion in 'paper explained' that tau and neurogranin can be used for monitoring is too hypothetical and not supported by the data, that do not include longitudinal sampling (only longitudinal clinical data).

***We have changed “monitor” to “detect” to make the conclusion compatible with our cross-sectional results (page 19, line 20).**

Although I do acknowledge the tremendous and important contribution of the Gothenburg studies and of the first author to the development of the field, it would be good if external citations are used more. There are places where these would have been at least equally good papers (e.g. use of NFL for monitoring neuroinflammation; Ng papers of Fagan group).

***We have added additional references for CSF NFL in relation to neuroinflammation (Christensen et al) and frontotemporal lobe dementia (Petzold et al). We cite the two papers on neurogranin from the Fagan group (Tarawneh et al, and Kester et al).**

AUC and AIC: abbreviations are not explained in the legend of table 3.

***The abbreviations are now explained.**

Furthermore, it would be better to have confidence intervals for the AUCs, which allows the reader to judge better the significance of the increase in AUC in extended models.

***We have added 95 % confidence intervals for the AUCs to the tables.**

Acceptance

08 July 2016

Please find enclosed the final report on your manuscript. We are pleased to inform you that your manuscript is accepted for publication and is being sent to our publisher to be included in the next available issue of EMBO Molecular Medicine.

Congratulations on your interesting work!

***** Reviewer's comments *****

Referee #2 (Comments on Novelty/Model System):

It is well done!

Referee #2 (Remarks):

Thanks for the careful explanations!

Corresponding Author Name: Niklas Mattsson

Manuscript Number: EMM-2016-06540